# Can Carbon Trading Policies Promote Regional Green Innovation Efficiency? Empirical Data from Pilot Regions in China

**Baoliu Liu, Zhenqing Sun * and Huanhuan Li**

School of Economics and Management, Tianjin University of Science & Technology, Tianjin 300457, China; 1748192699@mail.tust.edu.cn (B.L.); lhh9598@mail.tust.edu.cn (H.L.)

\* Correspondence: sunzq@tust.edu.cn

**Abstract:** Although the emission reduction and innovation effects of carbon emissions trading have attracted considerable interest among academics and policy makers, there is a lack of empirical research on how carbon trading pilots in China promote regional green innovation. Therefore, we measured the green innovation efficiency of 30 provinces and cities in mainland China from 2005 to 2018, using selected panel data within a super-efficient SBM model that incorporated undesirable outputs. We used a double differential model to evaluate the impacts of carbon trading policies on the green innovation efficiency of seven carbon trading pilot regions. These impacts were confirmed using the double differential propensity score matching method. Our findings were as follows. (1) The implementation of carbon trading policies has a significant and continuous effect of promoting and improving green innovation efficiency in the pilot areas. (2) Carbon trading induces technological innovation effects, enabling green innovation potential to be realized. Regional green innovation efficiency is further improved through energy substitution and structural upgrading effects and subsequently through all three of the above effects. (3) The synergy between the three major effects of carbon trading policies amplifies regional green innovation efficiency. Therefore, the Chinese government should accelerate its efforts to develop and improve carbon markets, promote carbon trading policies, and actively foster synergy among the three effects to achieve green and sustainable regional development.

**Keywords:** carbon trading; green innovation efficiency; propensity score matching; difference in differences model





## 1. Introduction

In recent years, while extensive economic growth has yielded economic dividends, it has also generated various problems, notably, excessive energy consumption and insufficient innovation. The question of how a win-win situation of economic growth and carbon dioxide ($CO_2$) emission reduction can be achieved, while making full use of essential resources, has emerged as a common societal concern. At the same time, the goal of establishing a global governance system for achieving green, low-carbon, and sustainable development is shared by countries globally. However, the task of achieving the reduction targets stipulated in the Paris Agreement for limiting the rise in the global temperature to no more than 2 °C above preindustrial levels and to reach the global peak as soon as possible is a daunting one [1]. In addition, through the implementation of large-scale carbon dioxide storage projects, the goals and requirements of the Paris Agreement can also be achieved. On the one hand, accelerated adjustment of the industrial structure to redress structural imbalances is required, and on the other hand, emission reduction technologies should be prioritized, and development should be propelled by innovation.

The question of whether a win-win situation for regional economic growth and carbon dioxide ($CO_2$) emission reduction can be achieved through the adoption of rational

economic policies and stringent environmental regulation is therefore a critical one. Efforts to explore and establish different carbon trading systems have been initiated in various regions worldwide. The European Union Emissions Trading Scheme is distinctive among these initiatives, as it is not only politically feasible but also environmentally effective, as well as cost-effective [2]. This system was rapidly extended to cover about 12,000 industrial and power facilities in Europe that were responsible for almost 50% of the EU's greenhouse gas emissions. Between 2005 and 2012, the EU-ETS accounted for 85% of the total global volume of carbon trading [3]. Within the United States, Chicago was the first city to participate in transactions to reduce greenhouse gas emissions, providing a foundation for national efforts to implement activities to reduce greenhouse gas emissions, such as RGGI, WCI, and the California Plan. Australia too launched a domestic carbon trading market after introducing relevant legislation in 2015, gradually developing a regional carbon market with extensive coverage, a carbon price compensation mechanism, and an improved monitoring mechanism over time [4].

In 2011, the Chinese government launched pilot carbon trading projects in seven provinces and cities, namely, Beijing, Tianjin, Shanghai, Chongqing, Guangdong, Hubei, and Shenzhen. The implementation of policies for establishing a carbon trading market in China has occurred relatively late compared with their implementation by other countries or organizations. Consequently, the carbon trading system is still evolving and requires further refinement, and there are information gaps on transactions. The Chinese government has adopted a series of market-oriented measures for achieving emission reduction targets through the development of a carbon trading system for promoting coordinated efforts to stimulate economic growth and environmental improvements. The government is simultaneously actively pursuing a path of green innovation and development framed through a series of concepts, notably, "innovative country", "wild China", and the "five concepts for development". "Green innovation" can also take diverse forms and include the promotion of economic growth [5]. Technical progress and environmental improvement initiatives in China are aimed at achieving the dual goals of "green mountains and clear water" and "mountains of gold and silver". Green innovation simultaneously entails a new process technology, system, and products aimed at reducing environmental pollution and damage and improving energy efficiency [6]. The goal of improving the efficiency of green innovation not only conforms to the concept of green development but it also encourages the implementation of innovation-driven development strategies.

To sum up, carbon trading has gradually gained popularity within most countries worldwide as a mechanism for promoting energy saving, emissions reduction, and low-carbon economic transformation through the use of market-oriented and mandatory measures to reduce energy consumption intensity. This approach also promotes $CO_2$ emissions reduction, which in turn drives the improvement of regional technological innovations. As the world's largest developing country and a major energy consumer, China's emission base is large, crucially impacting on the global carbon trading volume. Therefore, studies focusing on the operation of China's carbon trading mechanism and status quo would provide valuable inputs that could contribute to the realization of China's low-carbon economic transformation as well as the development of low-carbon technologies in other countries. The achievement of green and sustainable development of regional economies hinges on whether the technical effects of scale, and thus the level of regional technological innovation can be fully realized through the implementation of China's carbon trading policy. Using panel data extracted for 30 provinces and cities in mainland China for the period 2005–2018, we examined the green innovation potential of carbon trading. Specifically we explored the dynamic relationship between regional carbon emissions reduction and green innovation efficiency under the influence of China's carbon trading policy. Moreover, we sought to determine the mechanism by which carbon trading promotes regional green innovation efficiency. Elucidation of this mechanism contributes to advancing research on China's carbon trading regime and yields insights that can be applied to formulate guidelines for promoting green, high-quality regional development. Figure 1 shows the

distribution of China's pilot and non-pilot regions involved in the thesis research. Among them, the pilot regions mainly include the seven provinces and cities: Beijing, Tianjin, Shanghai, Hubei, Chongqing, Guangdong, and Shenzhen.

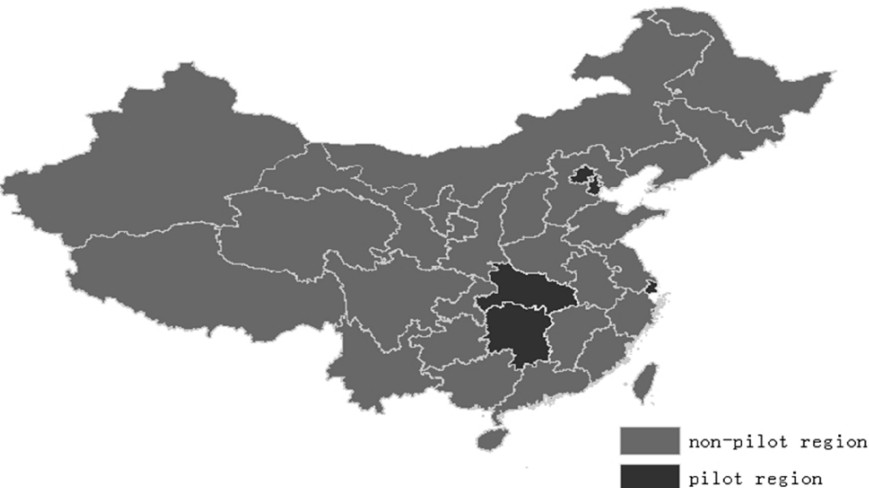

**Figure 1.** Comparative analysis of pilot and non-pilot regions.

## 2. Literature Review and Hypothesis

### 2.1. Literature Review

Scholars in China and abroad have conducted extensive research in the fields of carbon trading and green innovation. The research on carbon trading has mainly focused on impact effects, methods of measurement, carbon quotas, and carbon prices. This paper primarily focuses on studies on the impact effects of carbon trading. Dan et al. applied a differences-in-differences (DID) model to examine the policy effects of carbon trading. Their findings provided support for the "Porter hypothesis", revealing that the carbon trading mechanism encourages technological innovation to a certain extent. However, these authors did not find a strong policy effect on total factor productivity [7]. The trading mechanism of carbon emission rights is required to realize low-carbon transformation of the Chinese economy. Chuanming et al. applied a synthetic control method to investigate the carbon emissions reduction effect of China's carbon trading pilot provinces. They found that the carbon reduction effects of different pilot provinces were heterogeneous because of differences in their levels of economic development and in their industrial structures [8]. Other studies have examined the effects of carbon trading on economic growth. Zhengge et al. investigated whether the $SO_2$ emission trading mechanism exerts the Porter effect of inducing efficiency and innovation in China. They emphasized that strengthening market development and environmental regulation are necessary conditions for inducing the Porter effect [9]. Chunmei et al., who applied the directional distance function to calculate emissions reduction costs for China's industrial sector, confirmed that the carbon trading market has a significant impact on industries' emissions reduction costs and carbon intensity [10]. Studies that have explored synergistic effects include those of Cheng et al. and Ren Yayun. Their findings indicate that a carbon trading policy that encourages synergistic emissions reduction not only promotes the reduction of $CO_2$ emissions but it also promotes the reduction of other pollutants, thus playing a role in coordinated emissions reduction [11,12]. Last, a study by Jing and others examined the effect of upgrading the industrial structure, using the synthetic control method to evaluate the impacts of carbon trading on the upgrading of China's industrial structure. Its findings indicated that carbon trading compels the upgrading of the industrial structure [13].

Research on the efficiency of green innovation has primarily entailed the use of two approaches for measuring the values of efficiency and influencing factors. In the first approach, the parameter-based stochastic frontier model and the non-parametric data envelopment analysis methods are used to measure efficiency. In 1997, Chung and others

proposed the concept of a directional distance function, which provides methodological support for measuring total factor productivity, including "unexpected outputs" in a region. Watanabe et al. used the directional distance function to assess the impacts of China's inter-provincial initiative to eliminate bad industrial outputs on industrial efficiency during the period 1994–2002. They found that undesirable industrial outputs play an important role in improving industrial efficiency [14]. Neng et al. applied a hybrid DEA model to measure the efficiency of green innovation in China. The findings of their analysis of key factors affecting the efficiency of green innovation revealed that a good industrial structure, a free technology trading market, and basic environmental criteria had a greater impact on green innovation than did other factors [15]. Considering environmental pollution and innovation failure as undesirable outputs, Yanwei and others constructed an SBM–DEA model and alpha and beta convergence models to measure and converge the efficiency of green innovation in China [16]. The second approach entails the study of influencing factors. Scholars adopting this approach have largely focused on the following dimensions: the level of economic development [17], environmental regulation [18], R&D investments [19], and the level of interaction with the outside world [20]. These studies have advanced knowledge of the factors influencing green innovation efficiency within regions, while providing a conceptual foundation for initiatives aimed at improving levels of regional green innovation.

An examination of the recent literature reveals a paucity of studies that have investigated the relationship between carbon trading and green innovation efficiency and a lack of in-depth analysis of the internal mechanism driving this relationship. Moreover, few studies have examined the correlation between carbon emissions reduction and technological innovation despite the effectiveness of carbon trading, as a market-based policy tool, in reducing carbon intensity on the one hand and the importance of regional green innovation efficiency on the other hand. We aimed to address this gap by examining the green innovation potential of carbon trading. Our study makes the following contributions to the literature. First, it shows that the super-efficient SBM model, incorporating the impacts of undesirable outputs (carbon emissions), is an appropriate model for measuring green innovation efficiency in a region. Second, whereas several studies have examined the emissions reduction effects of carbon trading, they have not investigated the potential of carbon trading to promote regional green innovation. Therefore, we examined the diversity of regional green innovation efficiency within a carbon trading framework. Third, we argue that it is necessary to identify the intermediary transmission mechanism whereby carbon trading promotes improvements in a region's green innovation efficiency. At the same time, it is important to analyze differences in the technology innovation effects of carbon trading along with energy substitution and structural upgrading effects on green innovation efficiency. As we show, a study that entails both approaches can contribute useful insights for improving regional energy conservation and emission reduction capabilities and guiding inputs for the exploration of appropriate green innovation paths.

### 2.2. Research Hypothesis

As an environmental regulation tool, carbon trading influences the costs, benefits, and operating efficiency of the regional economy and promotes green, low-carbon regional development. Therefore, an in-depth study to examine the ways in which carbon trading contributes to the efficiency of green innovation within regions would shed light on their internal linkages.

(1) Technological innovation effect. Academic research has confirmed that technological innovation has a carbon emission reduction effect and that effective technological progress can significantly reduce carbon emissions [21,22]. Technological advances can help companies to control their emissions reduction costs and reduce production, thereby reducing carbon emissions. At the same time, companies located in pilot regions can use technological innovations to reduce emissions, and the technology spillover effect can further promote regional green innovation efficiency [12].

**Hypothesis 1 (H1).** *Carbon trading contributes to enhancing technological innovation in pilot areas, thereby increasing the level of regional green innovation efficiency.*

Energy substitution effect. The energy substitution effect of carbon trading is evidenced by increased production costs for traditional high-emission enterprises because of the need to expand their production scales. Moreover, $CO_2$ emission rights relating to ultra-carbon quotas have compelled enterprises to reduce their $CO_2$ emissions and increase the extent of their clean technology research and development [8]. Thus, the establishment of a carbon market can effectively stimulate the optimization and upgrading of the energy structures of enterprises, thereby increasing the proportion of clean energy that contributes to a reduction of carbon emissions. Clean energy is mainly used in the power sector, which has contributed more than 50% of carbon emissions [11]. Therefore, it is necessary to improve the energy utilization efficiency of this industrial sector through the prioritization of energy-saving and emission-reducing technologies and equipment that promote clean, low-carbon development in the pilot areas.

**Hypothesis 2 (H2).** *Carbon trading has led to the modification of the energy structure of enterprises in the pilot regions, and the substitution of fossil energy by clean energy has enabled the coordinated development of energy consumption and carbon emission reduction, thereby promoting regional green innovation efficiency.*

(2) Structural upgrading effect. An advanced industrial structure that is optimized and upgraded reflects the changing relationship between different industrial ratios and improved labor productivity within various industries and exerts "structural benefits" [23]. Carbon trading can contribute to the advancement of the industrial structure in two ways. The first entails changing the proportion of different industries and the second entails improving labor productivity within various industries [7]. At the same time, the optimization and upgrading of the industrial structure contributes to the flow of factors between industries, resulting in a gradual reduction in the proportion of the three industries associated with high carbon emissions, which leads to a reduction in industrial carbon emissions while simultaneously promoting green and low-carbon regional development.

**Hypothesis 3 (H3).** *Carbon trading drives the process of upgrading the industrial structure in the pilot regions, thereby improving the efficiency of regional green innovation.*

A carbon trading policy has three major effects that contribute to promoting green regional development: technological innovation, energy substitution, and structural upgrading effects. Figure 2 shows the specific mechanism whereby these effects are achieved.

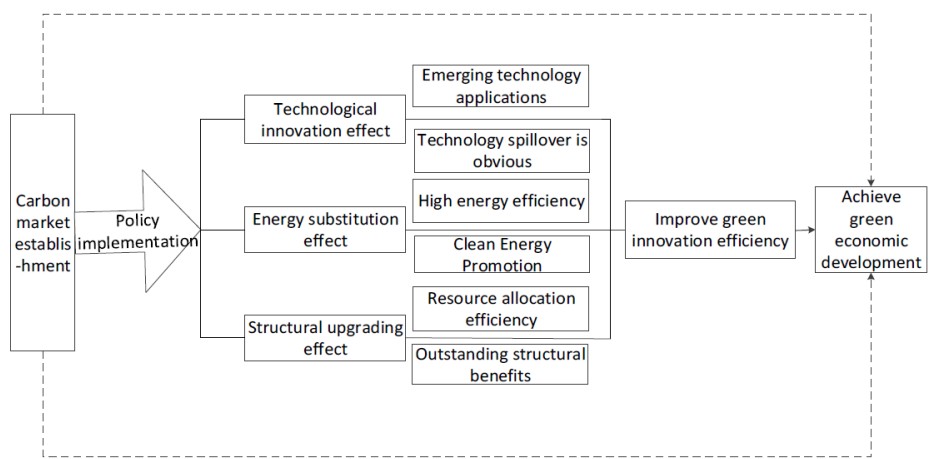

**Figure 2.** The mechanism whereby carbon trading policies influence the efficiency of green innovation.

## 3. Methodology and Models Applied

### 3.1. Super-Efficient SBM Model

Following our review of the relevant literature, we selected the non-parametric DEA method and the parametric SFA method for measuring green technology efficiency [24]. The DEA model can accommodate the relationship between multiple input and output variables. Moreover, it is more aligned with actual situations, as it does not require the setting of a specific function form. However, the traditional DEA model enables a unilateral assessment in which only the expected output is considered, while the effects of the undesirable output are ignored. Relaxation of inputs and outputs leads to a higher value for the measured efficiency. At the same time, the traditional DEA model is radial and angular, which results in some deviation in the calculated results. By contrast, non-radial and non-angle SBM models are better aligned with the study's requirements and address the above-mentioned issues. Consequently, we adopted the approach used in previous studies and incorporated undesirable outputs into the super-efficient SBM model [20,25]. We developed the following model for evaluating green innovation efficiency based on the assumptions that n kinds of decision-making units (DMUs) have m kinds of element inputs, $Z_1$ kinds of expected outputs, and $Z_2$ kinds of undesirable outputs:

$$\min \rho* = \frac{1 - \frac{1}{m}\sum\limits_{i=1}^{m}\frac{zi^-}{xik}}{1 + \frac{1}{z1+z2}\left(\sum\limits_{r=2}^{z1}\frac{z_r^g}{y_{rk}^g} + \sum\limits_{q=1}^{z2}\frac{z_q^b}{y_{qk}^g}\right)}$$

$$\text{s.t.} \begin{cases} xik = \sum\limits_{j=1}^{n} xij\lambda j + zi^- \ i = 1, \cdots, m \\ y_{rk}^g = \sum\limits_{j=1}^{n} yrj\lambda j - z_r^{g-} \ r = 1, \cdots, z1 \\ y_{qk}^g = \sum\limits_{j=1}^{n} y_{qj}^b\lambda j + z_q^{b-} \ q = 1, \cdots, z2 \end{cases} \tag{1}$$

$$\lambda j > 0 \ j = 1, \cdots, n$$
$$zi^- \geq 0, \ z_r^g \geq 0, \ z_q^b \geq 0$$

Model (1) indicates that when the invalid $DMU_k$ and the effective $DMU_k$ of $\rho*$ are transformed into each other, there is a corresponding correlation between a reduction in input variable $z_i^-$, an increase in the expected output $z_r^{g-}$, and a decrease in the undesirable output $z_q^{b-}$. When the values of $z^-$, $z^g$, and $z^b$ are larger, the efficiency value $\rho*$ of $DMU_k$ is correspondingly smaller, and when the values of $z^-$, $z^g$, and $z^b$ all have a value of 0, then $\rho* = 1$, indicating that $DMU_k$ is effective, there is no shortage in the expected output, and that the undesirable output is not in excess. $DMU_k$ in the super-efficient SBM model was deemed effective based on a consideration of the factors of the corresponding relaxation variables for the constraints. The specific model used for ranking DMUs was as follows:

$$\min \varphi* = \frac{1 - \frac{1}{m}\sum\limits_{i=1}^{m}\frac{\bar{x}}{xik}}{1 + \frac{1}{z1+z2}\left(\sum\limits_{r=2}^{z1}\frac{\bar{y}^g}{y_{rk}^g} + \sum\limits_{q=1}^{z2}\frac{s_q^b}{y_{qk}^g}\right)}$$

$$\text{s.t.} \begin{cases} \bar{x} \geq \sum\limits_{j=1,\neq k}^{n} xij\lambda j \ i = 1, \cdots, m \\ \bar{y}^g = \sum\limits_{j=1,\neq k}^{n} yrj\lambda j \ r = 1, \cdots, z1 \\ \bar{y}^b = \sum\limits_{j=1,\neq k}^{n} y_{qj}^b\lambda j \ q = 1, \cdots, z2 \end{cases} \tag{2}$$

$$\lambda j > 0 \ j = 1, \cdots, n \ j \neq 0$$
$$\bar{x} \geq xik, \ \bar{y}^g \leq y_{rj}^g, \ \bar{y}^b \leq y_{qj}^b$$

We derived and calculated model (2) on the basis of model (1). Using both of these models, we calculated the regional green innovation efficiency values of 30 provinces and cities in China (excluding the Tibet Autonomous Region) for the period 2005–2018 as follows:

$$GIE = \begin{cases} \theta *_{ik} & \theta *_{ik} < 1 \\ \omega *_{ik} & \omega *_{ik} = 1 \end{cases} \quad i = 1, \cdots 30, k = 2005, \cdots 2018 \tag{3}$$

where GIE denotes the green innovation efficiency value for area $i$ during year $k$.

### 3.2. DID Model

Currently, many methods exists for evaluating policy effects, such as synthetic control method and DID method [26]. In recent years, this DID model has been used for quantitative evaluations of public policy or for assessing project implementation effects within econometrics. The actual impacts of a policy can be assessed by comparing the amount of change for a specific indicator before and after the policy's implementation using an experimental group and a control group. The DID method has been widely used by scholars because of its ability to reduce endogeneity to some extent [26]. Therefore, we applied this model to investigate the regional emission reduction potential and the green innovation efficiency trend under conditions of the implementation of a carbon trading policy and to examine key factors that influence changes in efficiency.

There are currently seven pilot provinces and cities in China where the carbon trading policy has been implemented: Beijing, Tianjin, Chongqing, Shanghai, Hubei, Guangdong, and Shenzhen. To simplify the analysis, the city of Shenzhen was merged with Guangdong Province, and following Guangming et al., we set 2014 as the year demarcating the separation of the pilot and non-pilot periods, with the pilot period commencing from 2014 [27]. Accordingly, the following basic model was constructed:

$$GIEit = \alpha 0 + \alpha 1 Ci + \alpha 2 Yt + \alpha 3 (Ci \times Yt) + \lambda i + \gamma t + \mu it \tag{4}$$

where GIE denotes green innovation efficiency, $i$ denotes area, t denotes time, and $C_i$ denotes regional dummy variables. If province $i$ is a carbon trading pilot province or city, then the values of $C_i$ are 1 and 0, respectively, for the experimental and control groups. $Y_t$ denotes a time dummy variable. In 2014, which is the year of implementation of the policy, $t \geq 2014$ and $Y_t = 1$; otherwise 0. The estimated coefficients $\alpha 1$, $\alpha 2$, and $\alpha 3$ of the multiplication term $C_i \times Y_t$ are double-difference estimators, indicating the net impact of carbon trading policies. $\lambda_i$ denotes the individual fixed effects of provinces and cities, $\gamma_t$ denotes the fixed effect of time, and $\mu_{it}$ is the random interference term.

The explanatory use of model (4) on its own could result in the influence of other variables being discounted. Therefore, it is necessary to add control variables to the model to account for the influence of objective factors on the explanatory variables. With reference to previous studies, we transformed the basic model, selecting GDP per capita, R&D investment, carbon intensity, energy structure, population size, and R&D investment as the control variables:

$$GIEit = \alpha 0 + \alpha 1 Ci + \alpha 2 Yt + \alpha 3 (Ci \times Yt) + \sum \alpha j Xj + \lambda i + \gamma t + \mu it \tag{5}$$

where $X_j$ denotes the control variable, and the meaning of other variables is consistent with the above.

### 3.3. Selection of Variables and Data Sources

3.3.1. Interpreted Variables

Green innovation efficiency differs from other forms of innovation efficiency because it considers the impact of changes in energy consumption and carbon emissions on regional low-carbon development potential. Accordingly, drawing on the findings of previous studies that measured regional green innovation efficiency, we applied the super-efficient SBM model, incorporating undesirable outputs. Input elements, selected with reference

to the existing literature, were full-time R&D personnel, R&D funding inputs, and energy resource inputs.

Two categories of output elements of green innovation activities were defined: expected and undesirable outputs. Expected outputs were the value of economic growth, the number of authorized invention patent applications, and revenue from sales of new products. Undesirable outputs were innovation failures and the environmental pollution index. According to Schumpeter's definition of innovation, the success or failure of innovation is reflected in the generation (or not) of economic profits. Failure to innovate affects companies' ability to repay their business loans and their regular cash flows, making it impossible for them to generate profits. Consequently, they are left with non-performing loans. Accordingly, the undesirable output was calculated as the ratio of the amount of non-performing loans of commercial banks to the previous year. The environmental pollution index relates to the discharge of waste water, gas, and solid waste in various regions and was calculated using the entropy weight method to measure the weight of each indicator. The specific index system applied in this study is shown in Table 1.

**Table 1.** Evaluation index system used to measure green innovation efficiency.

| Index | Category | Index Composition | Specific Measurement |
|---|---|---|---|
| Input indicators | Factor input | R&D expenses | R&D expenditure (ten thousand yuan) |
| | | R&D staff | R&D personnel full-time equivalent (person, year) |
| | | Energy resources | Total energy consumption (10,000 tons of standard coal) |
| Output indicators | Expected output | The level of economic development | GDP per capita (ten thousand yuan, constant price in 2005) |
| | | Knowledge and technology output | Invention patent application authorization volume (pieces) |
| | | Product output | New product sales revenue (ten thousand yuan) |
| | Unexpected output | Innovation failure | Year-on-year ratio of non-performing loans of commercial banks (%) |
| | | Environmental Pollution Index | The entropy weight method is used to calculate the discharge of waste water, waste gas and solid waste |

### 3.3.2. Core Explanatory Variables

$C_i \times Y_t$ was the core explanatory variable. For a low-carbon city or province, when $Y \geq 2014$, the virtual variable $C_i \times Y_t$ corresponding to the city had a value of 1, and its coefficient indicated the net effect of the carbon trading policy and the strength of its emission reduction effect.

### 3.3.3. Control Variables and Measuring Indicators

Referring to the literature, we selected per capita GDP, R&D investments, carbon intensity, energy structure, and foreign capital dependence as the control variables relating to the level of regional green innovation efficiency.

### GDP Per Capita

In general, improvements in levels of regional economic development can drive technological innovation and enhance the level of low-carbon technological innovation. The regional GDP (constant price in 2005) and the proportion of permanent residents at the end of the year were used to express GDP per capita.

### R&D Investment Intensity

The level of scientific research has a crucial impact on the extent of regional technological innovation, thereby influencing energy usage. Regional R&D expenditure was used to reflect the amount of regional R&D investment.

Carbon Intensity

Carbon intensity was measured as the ratio of the total $CO_2$ emissions from fossil combustion in the region to the GDP.

Energy Structure

Coal is the main source of energy consumed. The ratio of total coal to total energy consumption was used to reflect changes in the energy structure, enabling a more intuitive understanding of the trend of energy ratio changes in various regions.

Foreign Capital Dependency

The amount of foreign direct investments significantly influences regional economic growth and the capabilities for controlling environmental pollution. Therefore, in this study, we considered the ratio of foreign direct investment to GDP to reflect the degree of foreign capital dependence.

### 3.4. Data Sources

Panel data were selected from the available raw data for 30 provinces in China (excluding Tibet, Hong Kong, Macau, and Taiwan) for the period 2005–2018. The data were extracted from the *China Statistical Yearbook*, the *China Energy Statistical Yearbook*, the *China Environmental Statistical Yearbook*, and the *China Science and Technology Statistical Yearbook*.

## 4. Empirical Results and Analysis

### 4.1. The DID Method of Regression Analysis

The DID method was applied in a further investigation of the effects of carbon trading policies on regional green innovation efficiency. Accordingly, we performed a regression analysis of the green innovation efficiency of 30 provinces and cities in China (excluding Tibet) for the period 2005–2018. In turn, the analysis was carried out for uncontrolled variables, using a two-way fixed-effect model that incorporates control variables, adds control variables, and controls regional and time effects, and analyzes the differences in impact under different circumstances.

Table 2 presents the results of an analysis to evaluate the impacts of carbon trading policies on green innovation efficiency using the DID method. Model (1) is a benchmark model for analyzing green innovation efficiency without control variables, whereas in model (2), control variables, such as per capita GDP, scientific research inputs, carbon intensity, foreign capital dependence, and energy structure were sequentially added. Model (3) is a model with time effects added to model (2). Overall, with the increase in control variables, the significance of the core explanatory variables and the signs of the coefficients did not change appreciably, and were significantly positive at the 5% level, indicating that the model results were robust. The impact of GDP per capita on green innovation efficiency relating to the control variable changed from negative to positive at a 10% significance level. This result indicates that after the implementation of the carbon trading policy, the pilot regions paid more attention to the coordinated development of economic growth and environmental protection, thereby promoting improvements in regional levels of green innovation. R&D investment was positive at a 1% significance level, indicating that regional R&D investments play a definitive role in promoting technological innovation. The impacts of carbon intensity and the energy structure on the efficiency of green innovation changed from negative to positive, indicating that the reduction effect of carbon trading policies propelled the energy structure's optimization and enhanced green innovation efficiency within a region. Dependence on foreign capital had a positive effect on the efficiency of green innovation, but this effect was not significant. This finding indicates the importance of expanding the proportion of foreign investment and improving the level of technological progress to improve the level of green innovation in a region.

**Table 2.** Regression results for the impacts of carbon trading policies on green innovation efficiency.

| Variable | Green Innovation Efficiency | | |
|---|---|---|---|
| | (1) | (2) | (3) |
| Ci × Yt | 0.1332 ** (2.39) | 0.0899 ** (2.01) | 0.0498 *** (3.45) |
| GDP per capita | | −0.3653 * (−1.73) | 0.4929 * (1.86) |
| Research investment | | 0.3692 *** (4.62) | 0.3678 *** (4.27) |
| Carbon intensity | | −0.4379 * (−1.94) | 0.6479 ** (2.17) |
| Foreign capital dependency | | 0.0525 (1.29) | 0.0635 (1.42) |
| energy structure | | −0.1103 (−0.43) | 0.0509 * (1.80) |
| Control variable | NO | YES | YES |
| Province fixed | YES | YES | YES |
| Fixed year | NO | NO | YES |
| Constant term | 0.1756 *** (8.10) | −1.2177 *** (−4.54) | −2.6662 *** (−6.44) |
| N | 420 | 420 | 420 |
| $R^2$ | 0.0374 | 0.3850 | 0.4274 |

Notes: *t*-values are shown in brackets; ***, **, and * indicate statistical significance at the 1%, 5%, and 10% levels, respectively.

### 4.2. Analysis Using the PSM Method

The DID method must meet the requirements of the parallel trend assumption, and whether to implement the carbon trading policy as a virtual variable for overall regression, the parameters may be biased. Therefore, the PSM–DID method was used for the estimation. As required for the PSM method, the Logit model was used to estimate the propensity scores of per capita GDP, scientific research investment, carbon intensity, foreign capital dependence, and the energy structure. These scores are shown in Table 3. Subsequently, provinces within the experimental and control groups were matched using the kernel matching method. At the end of this procedure, 10 samples that did not match the cost were deleted. Regression analysis was performed again using the matching data, and the results are shown in Table 4.

**Table 3.** Logit regression estimation results using the PSM method.

| Variable | Coefficient | Standard Error | T | P |
|---|---|---|---|---|
| GDP per capita | 0.2843 ** | 0.1435 | 1.98 | 0.021 |
| Research investment | 0.3737 *** | 0.1341 | 2.79 | 0.005 |
| Carbon intensity | −0.0215 *** | 0.0070 | −3.06 | 0.002 |
| Foreign capital dependency | 0.0062 ** | 0.0002 | 2.14 | 0.030 |
| energy structure | −0.7002 *** | 0.2574 | −2.72 | 0.003 |
| -cons | 1.8016 *** | 0.6656 | 2.71 | 0.000 |

Notes: *** and ** indicate statistical significance at the 1%, 5%, and 10% levels, respectively.

Table 3 shows estimation results for the control variables obtained using the Logit regression model. The significance level was high for each variable, which is consistent with the actual situation. Advanced regional economic development along with increased proportions of R&D investments and emissions reduction efforts were indicative of an increased proportion of foreign direct investment and of an optimized and upgraded energy structure. Instead, it is easier to enter the experimental group to ensure the reliability of the regression results. Table 4 depicts the results of continued testing of the carbon trading policy using the DID after deleting unmatched samples. The regression results reveal that overall, the matching carbon trading policy resulted in enhanced green innovation efficiency, which increased by 0.01 units. The increase in GDP per capita was also caused by changes in the matched samples, and the economic dividend phenomenon was clearly apparent, contributing significantly to enhanced green innovation efficiency. At the same time, investments in scientific research, reduced carbon intensity, and green innovation efficiency within a region's energy structure evidently had positive effects. Moreover, with the implementation of the carbon trading policy, this effect was amplified. Thus, the question of whether a mechanism whereby carbon trading policies influence green innovation

efficiency in the pilot areas exists, and, if so, how it operates, required an in-depth analysis. This investigation to elucidate and verify such a mechanism is described next.

**Table 4.** Regression results showing the role of carbon trading policies before and after the propensity score matching process.

| | Green Innovation Efficiency | | | | | |
| --- | --- | --- | --- | --- | --- | --- |
| | **Before Matching** | **After Matching** | **Before Matching** | **After Matching** | **Before Matching** | **After Matching** |
| $C_i \times Y_t$ | 0.1332 ** (2.39) | 0.2063 *** (2.67) | 0.0899 ** (2.01) | 0.0149 *** (3.11) | 0.0498 *** (3.45) | 0.0676 ** (2.22) |
| GDP per capita | | | −0.3653 * (−1.73) | 0.0891* (1.91) | 0.4929 * (1.86) | 0.2844 ** (2.30) |
| Research investment | | | 0.3692 *** (4.62) | 0.4073 *** (3.44) | 0.3678 *** (4.27) | 0.3737 ** (3.79) |
| Carbon intensity | | | −0.4379 * (−1.94) | −0.2137 (−0.01) | 0.6479 ** (2.17) | 0.0215 ** (2.06) |
| Foreign capital dependency | | | 0.0525 (1.29) | 0.0006 * (1.81) | 0.0635 (1.42) | 0.0063 * (1.92) |
| energy structure | | | −0.1103 (−0.43) | 0.7981 (1.04) | 0.0509 * (1.80) | 0.7002 ** (2.32) |
| Control variable | NO | NO | YES | YES | YES | YES |
| Province fixed | YES | YES | YES | YES | YES | YES |
| Fixed year | NO | NO | NO | NO | YES | YES |
| Constant term | 0.1756 *** (8.10) | 0.2558 *** (6.35) | −1.2177 *** (−4.54) | −1.1822 *** (−2.76) | −2.6662 *** (−6.44) | −1.8016 *** (−2.71) |
| N | 420 | 260 | 420 | 260 | 420 | 260 |
| $R^2$ | 0.4274 | 0.8901 | 0.3850 | 0.9225 | 0.4274 | 0.3532 |

Notes: *t* values are shown in brackets; ***, **, and * indicate statistical significance at the 1%, 5%, and 10% levels, respectively.

*4.3. Testing and Verification of an Intermediary Influence Mechanism*

The above empirical results reveal that the pilot provinces and cities where carbon trading policies have been implemented could significantly improve their levels of green innovation efficiency. However, a macroscale analysis of the impact of carbon trading policies on green innovation efficiency lacks sufficient depth for exploring the impact mechanism behind the policy effect. As previous research has shown, carbon trading policies have a significant role to play in promoting low-carbon development of provinces and cities, which, in turn, propels technological innovation, energy substitution, and structural upgrading and improves the efficiency of green innovation. To elucidate the impact mechanism and verify the existence of these three effects, the following steps were implemented, applying the formulas described by Baron and Kenny and Daqian [28,29].

The first step entailed verification of the three major effects of the carbon trading policy in pilot regions using the following formula:

$$TI_{it}(ES_{it}, SU_{it}) = \beta 0 + \beta 1 C_i + \beta 2 Y_t + \beta 3 (C_i \times Y_t) + \sum \beta j X_j + \lambda i + \gamma t + \mu it \quad (6)$$

The second step entailed verifying the impacts of carbon trading policies on green innovation efficiency as follows:

$$GIE_{it} = \beta 0 + \beta 1 C_i + \beta 2 Y_t + \beta 3 (C_i \times Y_t) + \sum \beta j X_j + \lambda i + \gamma t + \mu it \quad (7)$$

In the third step, the multiplier term and the three major effects were simultaneously inputted into the model and returned to the green innovation efficiency:

$$GIE_{it} = \theta 0 + \theta 1 C_i + \theta 2 Y_t + \theta 3 (C_i \times Y_t) + \theta 4 TI_{it}(ES_{it}, SU_{it}) \sum \theta j X_j + \lambda i + \gamma t + \mu it \quad (8)$$

where TI denotes the effect of technological innovation, expressed by the number of patent applications for energy conservation and emission reduction technologies in various

regions. The specific data for the calculation, which were acquired using the method described by Ye Qin et al., indicated that carbon trading enhances the level of regional technological innovation, inducing a technological innovation effect and improving the efficiency of green innovations [30]. ES is the energy substitution effect, entailing the replacement of a proportion of the total amount of regional electricity that is consumed by clean energy. A higher proportion of clean energy used as a substitute within a region corresponds to a stronger effect of enhanced green development. SU is the structural upgrading effect, expressed as the ratio of the tertiary industry to secondary industry within a region. Carbon trading evidently promotes the advancement of the regional industrial structure, and this structural upgrading effect improves the level of green innovation.

Table 5 shows the regression results for the impacts of carbon trading policies relating to the three major effects. The results show that the regression coefficients of the three major effects were all positive at the 1% significance level, indicating that carbon trading policies influence technological innovation, energy substitution, and structural upgrading in the process of applying market-oriented strategies to achieve emission reduction goals. Table 6 shows the regression results for the impacts of carbon trading policies on green innovation efficiency after including the difference factors. The results indicate that the regression coefficients of the effects of technological innovation, energy substitution, and structural upgrading were significantly positive. Specifically, the effect of technological innovation on green innovation efficiency was positive at a 1% significance level, highlighting the important role of the development and use of energy-saving and emission-reducing technologies for advancing green and sustainable regional development. The effects of energy substitution and structural upgrading on green innovation efficiency were positive at the 10% and 5% significance levels, indicating that both of these effects play a role in promoting efficient green innovation, but these effects are not particularly significant. Moreover, the results indicate that China's current energy structure continues to be unviable, as it is still overly dependent on traditional fossil energy sources and requires further optimization and upgrading. These results also confirm the three hypotheses of the study, namely that carbon trading policies improve the efficiency of green innovation within regions through their effects on technological innovation, energy substitution, and structural upgrading.

**Table 5.** The regression results for carbon trading policies on the three major effects.

| Variable | Technological Innovation Effect | Energy Substitution Effect | Structural Upgrading Effect |
|---|---|---|---|
| $C_i \times Y_t$ | 0.0162 *** (3.13) | 0.2025 *** (5.21) | 0.0266 *** (4.33) |
| N | 420 | 420 | 420 |
| $R^2$ | 0.8043 | 0.9012 | 0.8624 |

Notes: *t* values are shown in brackets; *** indicate statistical significance at the 1%, 5%, and 10% levels, respectively.

**Table 6.** The regression results for carbon trading policies on green innovation efficiency after adding the multiple differences.

| Variable | Green Innovation Efficiency | | |
|---|---|---|---|
| $C_i \times Y_t$ | 0.2308 *** (3.12) | 0.0923 ** (2.10) | 0.0831 *** (4.02) |
| Technological innovation | 0.3425 *** (2.90) | | |
| Energy substitution | | 0.0565 * (1.93) | |
| Structural upgrade | | | 0.2602 ** (2.08) |
| N | 420 | 420 | 420 |
| $R^2$ | 0.8913 | 0.8265 | 0.9031 |

Notes: *t* values are shown in brackets; ***, **, and * indicate statistical significance at the 1%, 5%, and 10% levels, respectively.

### 4.4. Robustness Test for Changing the Sample Interval

The implementation effect of carbon trading policy needs time to test. Considering the timeliness issues before and after the policy, a more balanced data sample for the period

2010–2018 was selected for a further regression conducted to test the robustness of the main regression.

Table 7 shows the regression results of the robustness test during the change sample interval. Columns (1), (2), and (3) show the results of the carbon trading policy on the regressions for the technological innovation, energy substitution, and structural upgrading effects, and column (4) shows the regression results for the three major effects and for green innovation efficiency. The regression results indicate that the coefficients of the interaction terms were all significant at the 1% and 5% levels, and the coefficients of the three major effects were also significant at the 5% and 10% levels. These findings are consistent with those of the main regression, described above, confirming that the regression results presented in this paper are robust. Thus, the findings indicate that the carbon trading policy has produced three major effects and that it has enhanced regional green innovation efficiency through the mechanism of these three effects.

**Table 7.** Results of the regression to test the robustness of the main regression and to change the window period.

| Variable | Technological Innovation Effect | Energy Substitution Effect | Structural Upgrading Effect | Green Innovation Efficiency |
|---|---|---|---|---|
| | (1) | (2) | (3) | (4) |
| $C_i \times Y_t$ | 0.5566 *** (5.78) | 0.2119 ** (2.56) | 0.2572 *** (8.04) | |
| Technological innovation effect | | | | 0.2522 ** (2.27) |
| Energy substitution effect | | | | 0.1051 * (1.90) |
| Structural upgrading effect | | | | 0.1296 * (1.78) |
| N | 270 | 270 | 270 | 270 |
| $R^2$ | 0.7890 | 0.8248 | 0.8932 | 0.9023 |

Notes: *t* values are shown in brackets; ***, **, and * indicate statistical significance at the 1%, 5%, and 10% levels, respectively.

## 5. Conclusions and Policy Implications

### 5.1. Conclusions

Panel data obtained for 30 provinces and cities in China (excluding Tibet) were applied within a super-efficient SBM model that included undesirable outputs to measure these regions' green innovation efficiency for the period 2005–2018. A dual difference model and the PSM–DID method were simultaneously used in an empirical examination of the trend in regional green innovation efficiency as it has been impacted by the implementation of carbon trading policies. The conclusions of the study can be summarized as follows.

First, the implementation of carbon trading policies can significantly improve the efficiency of green innovation in pilot areas and promote green, low-carbon regional development. Second, we investigated the influence mechanism of carbon trading policies as a driver of increased efficiency of green innovation. Our findings indicated that the implementation of carbon trading policies improves the efficiency of green innovation through its effects relating to technological innovation, energy substitution, and structural upgrading. The technological innovation effect was positively significant at the 1% level, while the effects of energy substitution and structural upgrading were positively significant, though only at the 10% and 5% levels. Last, our empirical findings revealed the overall synergistic effect of the three individual effects of carbon trading policies in amplifying regional green innovation efficiency.

### 5.2. Policy Implications

In light of the above conclusions, our findings have the following policy implications for improving the development of China's carbon market. First, the successful experiences in the pilot regions can be replicated by promoting the carbon trading policy on a wider scale and through the advancement of the national carbon market. The implementation of carbon trading policies can play a major role in reducing carbon emissions and driving transformational, low-carbon regional development. On the one hand, further exploration of low-carbon development is needed in the carbon trading pilot areas, focusing on

technological innovation and the development of greener production and lifestyles that incorporate, for example, green transportation, green buildings, and green consumption. On the other hand, further development of the pilot regions would enable the expanded scope of market transactions in terms of points and areas and the use of market-oriented strategies to compel companies to upgrade their levels of technological innovation, thereby promoting China's overall green development.

Second, continued efforts are required to increase the proportion of R&D investment in technology, while simultaneously increasing the proportion of clean energy use and adjusting and upgrading the industrial structure. Specifically, the patent incentive system requires improvement, and enterprises or innovation-focused entities should be provided with guidance. Such guidance should focus on the invention and application of energy-saving and emission-reduction patented technologies, reducing the costs for enterprises of investing in new technologies, and providing innovation subsidies for carbon quotas to facilitate green production processes. Further, the process of transforming the energy structure to promote improved green innovation efficiency should be accelerated along with the development of new clean energy and efforts to accelerate reforms of the electricity market system to promote the greening of the power generation process. Finally, efforts to increase the proportion of high-quality service industries, while simultaneously focusing on rationalizing and gradually optimizing the industrial structure, would contribute to its advancement and ensure a gradual rise in levels of regional green innovation.

A final implication of this study relates to the need to develop an institutional system and mechanism for promoting low-carbon development, building on the synergistic effects of technological innovation, energy substitution, and structural upgrades under carbon trading policies. The development of green innovations within regions requires the promotion of advanced green technology and industrial upgrading within various industrial sectors. It simultaneously requires innovations within the macro-control energy system that facilitate the phasing out of energy-intensive and carbon-intensive industries, and the dismantling of the "locking effect" of energy-driven development. By ensuring the coordinated and balanced development of different policy effects, this approach will lead to enhanced efficiency of green innovation within regions and high-quality economic development.

**Author Contributions:** Conceptualization, B.L. and Z.S.; methodology, H.L.; software, H.L.; validation, B.L., Z.S. and H.L.; formal analysis, B.L.; investigation, H.L.; resources, H.L.; data curation, H.L.; writing—original draft preparation, B.L.; writing—review and editing, H.L.; visualization, Z.S.; supervision, Z.S.; project administration, Z.S.; funding acquisition, Z.S. All authors have read and agreed to the published version of the manuscript.

**Funding:** This research was funded by the National Social Science Foundation of China, grant number (16AGL002); the Ministry of Education Philosophy and Social Sciences Research Fund, grant number (16JZD014); the Tianjin College Innovation Team Training Program, grant number (TD13-5012/5045).

**Institutional Review Board Statement:** Ethical review and approval was not required for the study as the research does not involve humans.

**Informed Consent Statement:** Not applicable.

**Data Availability Statement:** Data available on request.

**Conflicts of Interest:** The authors declare no conflict of interest.

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
