# Peer review of "Can Carbon Trading Policies Promote Regional Green Innovation Efficiency? Empirical Data from Pilot Regions in China"

_sustainability, doi:10.3390/su13052891_

Round 1

Reviewer 1 Report

The title of the article suggests an attractive and current issue in the trends of world science. The title, however, indicates a regional approach, which was not emphasised in the text of the article. Firstly, there is no presentation of pilot regions mentioned in the title and summary. And yet even the best calculations should refer to the space they concern. There is a lack of characteristics of differences between the studied regions and a summary of results and conclusions in regional perspective. Does the applicability of regional policy translate into indicators in different regions? Unfortunately, the article does not answer this question.

Apart from showing the regional specifics of the comparisons obtained, the usual presentation of the location of the regions in discussion against the map of China is missing. This is desirable in international publications to make the reader's perception easier also outside China. The list of cited literature also conspicuously lacks a cross-sectional world context on the various approaches to green innovations.

Author Response

Point 1: The title of the article suggests an attractive and current issue in the trends of world science. The title, however, indicates a regional approach, which was not emphasised in the text of the article. Firstly, there is no presentation of pilot regions mentioned in the title and summary. And yet even the best calculations should refer to the space they concern. There is a lack of characteristics of differences between the studied regions and a summary of results and conclusions in regional perspective. Does the applicability of regional policy translate into indicators in different regions? Unfortunately, the article does not answer this question.

Response to comment:

Dear Reviewer:

Thank you for your letter and for the reviewers comments concerning our manuscipt. These comments are all valuable and very helpful for revising and improving our paper, as well as the important guiding significance to our researchs. We have studied comments carefully and have made correction which we hope meet with approval.

       Thank you for your suggestions. In response to your regional issues, the topic of the thesis is to explore the impact of the implementation of carbon trading policies on the efficiency of regional green innovation, and the empirical content of the article is simple in the early stage of the pilot and non-pilot areas. For comparative analysis, there is no specific quantification to analyze the characteristics of different regions. The paper mainly focuses on the overall consideration, based on the panel data of 30 provinces and cities in China, and analyzes it from the macro level. This is also where the article lacks further elaboration and will be further analyzed in the following research.

Point 2:Apart from showing the regional specifics of the comparisons obtained, the usual presentation of the location of the regions in discussion against the map of China is missing. This is desirable in international publications to make the reader's perception easier also outside China. The list of cited literature also conspicuously lacks a cross-sectional world context on the various approaches to green innovations.

Response to comment:

Thank you for your suggestions. Considering that there may be problems with unclear borders when using Chinese maps, this part of the display is avoided in the paper. At the same time, the references in the thesis were revised, and more research documents on green innovation efficiency were cited to improve the corresponding conclusions.

Special thanks to you for your good comments

Reviewer 2 Report

This is a very informative paper including valuable long-term data. The methods, hypothesis, results, and conclusions are written clearly. I wish to suggest the authors provide a clear statement in both abstract and introduction to clarify the objectives of the work. There are also minor lingual issues that can be resolved by letting the work undergo English language editing.

Author Response

Point 1: This is a very informative paper including valuable long-term data. The methods, hypothesis, results, and conclusions are written clearly. I wish to suggest the authors provide a clear statement in both abstract and introduction to clarify the objectives of the work. There are also minor lingual issues that can be resolved by letting the work undergo English language editing.

Response to comment:

Dear Reviewer:

Thank you for your letter and for the reviewers comments concerning our manuscipt. These comments are all valuable and very helpful for revising and improving our paper, as well as the important guiding significance to our researchs. We have studied comments carefully and have made correction which we hope meet with approval.

Thanks for your advice. In response to your proposed methods, hypotheses, results and conclusions, we have carefully considered your suggestions and revised and improved the article. At the same time, the expressions in the abstract and introduction have been improved to clearly reflect the goals of the work. In addition, we also contacted the English editor to resolve the language issues in this article.

Special thanks to you for your good comments.

Reviewer 3 Report

This paper evaluates the impacts of carbon trading policies on the green innovation efficiency in China. The findings can have implications in sustainable development of green technologies. This is a nice paper and need some modifications as noted below:

  • Line 53: please note that beside the mentioned objectives, large scale CO2 storage projects should also be implemented to achieve the objectives of Paris agreement. See the following articles for more detail: org/10.1021/acs.est.7b05784 , doi.org/10.1021/acssuschemeng.8b06374
  • Introduction: the story line in the introduction is strong. However, it reads like report more than a scientific article. I would strongly suggest to include studies of similar kind in the introduction.
  • Literature review: this part is better to be combined with the introduction.
  • Literature review: it would be good to include studies on other region of the world.
  •  2. Research hypothesis: many recent works are missing in this part. There are also lots of sentences without references.
  • 2. DID model: authors mentioned “many methods exists for evaluating policy effects, one of which is the DID method”: authors should provide brief information about other models or should provide a reference that summarized other methods.
  • Line 301: how authors estimate the following variables: . Please provide more detailed information in the article
  • 3.3. Control variables and measuring indicators: authors mentioned “Referring to the literature, we selected per …” they should cite the reference from the literature they referring to…
  • Empirical results and analysis: authors should explain why they chose 1%, 5%, and 10% significance levels
  • Line 514: authors mentioned “a more balanced data samples” could the authors explain what they mean by “a more balanced”?
  • Results and discussion: most of the conclusions made by the authors have already been mentioned in other research for other countries. This would be a positive point for your article if you mention those conclusion which are in agreement or in contrast with your results.
  • Authors mentioned “A final implication of this study relates to the need to develop an institutional system and mechanism for promoting low-carbon development, building on the synergistic effects of technological innovation, energy substitution, and structural upgrades under carbon trading policies.” The statement is true and well stablished. However, it is not an implication of this study. There are other similar statements in the article. Please use more balanced words and give a credit for the people who developed the field.

Author Response

Point 1:Line 53: please note that beside the mentioned objectives, large scale CO2 storage projects should also be implemented to achieve the objectives of Paris agreement. See the following articles for more detail: org/10.1021/acs.est.7b05784 , doi.org/10.1021/acssuschemeng.8b06374

Response to comment:

Dear Reviewer:

Thank you for your letter and for the reviewers comments concerning our manuscipt. These comments are all valuable and very helpful for revising and improving our paper, as well as the important guiding significance to our researchs. We have studied comments carefully and have made correction which we hope meet with approval.

Thank you for your suggestion. The content of line 53 has been modified according to your reference information.

Point 2:Introduction: the story line in the introduction is strong. However, it reads like report more than a scientific article. I would strongly suggest to include studies of similar kind in the introduction.

Response to comment: According to  your suggestions, we have improved the introduction to ensure the logic of the sentence.

Point 3:Literature review: this part is better to be combined with the introduction.

Literature review: it would be good to include studies on other region of the world.

Response to comment: According to your suggestions, we will combine the literature review part with the introduction, and add relevant research from other countries.

Point 4: Research hypothesis: many recent works are missing in this part. There are also lots of sentences without references.

Response to comment: According to your suggestions, we revised the research hypothesis of the paper and added the references.

Point 5: DID model: authors mentioned “many methods exists for evaluating policy effects, one of which is the DID method”: authors should provide brief information about other models or should provide a reference that summarized other methods.

Response to comment: According to  your suggestions, we have added references to other model methods.

Point 6: Line 301: how authors estimate the following variables: . Please provide more detailed information in the article

Response to comment: According to  your suggestions, we have supplemented the relevant variables in the paper.

Point 7: Control variables and measuring indicators: authors mentioned “Referring to the literature, we selected per …” they should cite the reference from the literature they referring to…

Empirical results and analysis: authors should explain why they chose 1%, 5%, and 10% significance levels

Response to comment:According to  your suggestions, we added references and explained the significance coefficient.

Point 8: Line 514: authors mentioned “a more balanced data samples” could the authors explain what they mean by “a more balanced”?

Response to comment:Selecting more balanced sample data in the paper is mainly used for robustness analysis. Considering that the volatility of sample data in some years is relatively large, the impact on the results is relatively large. Therefore, the 2010-2018 data is selected for the robustness test in the paper.

Point 9: Results and discussion: most of the conclusions made by the authors have already been mentioned in other research for other countries. This would be a positive point for your article if you mention those conclusion which are in agreement or in contrast with your results.

Response to comment:According to  your suggestions, we try to add new conclusions to enrich the research results of the thesis.

Point 10:Authors mentioned “A final implication of this study relates to the need to develop an institutional system and mechanism for promoting low-carbon development, building on the synergistic effects of technological innovation, energy substitution, and structural upgrades under carbon trading policies.” The statement is true and well stablished. However, it is not an implication of this study. There are other similar statements in the article. Please use more balanced words and give a credit for the people who developed the field.

Response to comment:According to your suggestions, we have improved the corresponding language expression to ensure the rigor of the paper.

Special thanks to you for your good comments.

Round 2

Reviewer 1 Report

Dear authors, the changes you have made at my suggestion are cosmetic. I am persuaded by the argument about the macro level, but if it is accompanied by a regional context, it is no longer a macro level. The macro level should refer to China as a whole and its results. If each region is treated separately, then we have a regional take. It has always been the custom in international journals to show the location of research areas on maps. This custom serves to improve the perception of the issues presented. Even if you do not have compiled data assigned to a region, it should not be a big problem nowadays to show the location of the analysed regions on a map. I strongly suggest showing the location of the regions considered in the study graphically. Spatial context is important for international presentation.

Author Response

Point 1: Dear authors, the changes you have made at my suggestion are cosmetic. I am persuaded by the argument about the macro level, but if it is accompanied by a regional context, it is no longer a macro level. The macro level should refer to China as a whole and its results. If each region is treated separately, then we have a regional take. It has always been the custom in international journals to show the location of research areas on maps. This custom serves to improve the perception of the issues presented. Even if you do not have compiled data assigned to a region, it should not be a big problem nowadays to show the location of the analysed regions on a map. I strongly suggest showing the location of the regions considered in the study graphically. Spatial context is important for international presentation.

Response to comment:

Dear Reviewer:

Thank you for your letter and for the reviewers comments concerning our manuscipt. These comments are all valuable and very helpful for revising and improving our paper, as well as the important guiding significance to our researchs. We have studied comments carefully and have made correction which we hope meet with approval.

      Thank you for your suggestions. According to your suggestions, we have added a description of the pilot and non-pilot regionss involved in the thesis research. By adopting the regional approach, the specific regiona studied is displayed spatially, and the specific regions involved is shown in the figure below.
